# Characteristic and Chondrogenic Differentiation Analysis of Hybrid Hydrogels Comprised of Hyaluronic Acid Methacryloyl (HAMA), Gelatin Methacryloyl (GelMA), and the Acrylate-Functionalized Nano-Silica Crosslinker

**DOI:** 10.3390/polym14102003

**Published:** 2022-05-13

**Authors:** Swathi Nedunchezian, Che-Wei Wu, Shung-Cheng Wu, Chung-Hwan Chen, Je-Ken Chang, Chih-Kuang Wang

**Affiliations:** 1Department of Medicinal and Applied Chemistry, Kaohsiung Medical University, Kaohsiung 80701, Taiwan; sofiyaswathy@gmail.com; 2Regenerative Medicine and Cell Therapy Research Center, Kaohsiung Medical University, Kaohsiung 80701, Taiwan; tkdiven@gmail.com (C.-W.W.); shunchengwu@gmail.com (S.-C.W.); hwan@kmu.edu.tw (C.-H.C.); jkchang@kmu.edu.tw (J.-K.C.); 3Orthopedic Research Center, College of Medicine, Kaohsiung Medical University, Kaohsiung 80701, Taiwan; 4Department of Orthopedics, College of Medicine, Kaohsiung Medical University, Kaohsiung 80701, Taiwan; 5Department of Orthopedics, Kaohsiung Municipal Ta-Tung Hospital, Kaohsiung Medical University, Kaohsiung 80701, Taiwan; 6Division of Adult Reconstruction Surgery, Department of Orthopedics, Kaohsiung Medical University Hospital, Kaohsiung Medical University, Kaohsiung 80701, Taiwan; 7Graduate Institute of Medicine, College of Medicine, Kaohsiung Medical University, Kaohsiung 80701, Taiwan

**Keywords:** hyaluronic acid, gelatin, hybrid hydrogel, inorganic crosslinker, photocrosslinking, adipose stem cell, chondrogenesis

## Abstract

Developing a biomaterial suitable for adipose-derived stem cell (ADSCs)-laden scaffolds that can directly bond to cartilage tissue surfaces in tissue engineering has still been a significant challenge. The bioinspired hybrid hydrogel approaches based on hyaluronic acid methacryloyl (HAMA) and gelatin methacryloyl (GelMA) appear to have more promise. Herein, we report the cartilage tissue engineering application of a novel photocured hybrid hydrogel system comprising HAMA, GelMA, and 0~1.0% (*w*/*v*) acrylate-functionalized nano-silica (AFnSi) crosslinker, in addition to describing the preparation of related HAMA, GelMA, and AFnSi materials and confirming their related chemical evidence. The study also examines the physicochemical characteristics of these hybrid hydrogels, including swelling behavior, morphological conformation, mechanical properties, and biodegradation. To further investigate cell viability and chondrogenic differentiation, the hADSCs were loaded with a two-to-one ratio of the HAMA-GelMA (HG) hybrid hydrogel with 0~1.0% (*w*/*v*) AFnSi crosslinker to examine the process of optimal chondrogenic development. Results showed that the morphological microstructure, mechanical properties, and longer degradation time of the HG+0.5% (*w*/*v*) AFnSi hydrogel demonstrated the acellular novel matrix was optimal to support hADSCs differentiation. In other words, the in vitro experimental results showed that hADSCs laden in the photocured hybrid hydrogel of HG+0.5% (*w*/*v*) AFnSi not only significantly increased chondrogenic marker gene expressions such as SOX-9, aggrecan, and type II collagen expression compared to the HA and HG groups, but also enhanced the expression of sulfated glycosaminoglycan (sGAG) and type II collagen formation. We have concluded that the photocured hybrid hydrogel of HG+0.5% (*w*/*v*) AFnSi will provide a suitable environment for articular cartilage tissue engineering applications.

## 1. Introduction

Articular cartilage damage is a common joint disease that affects millions of people worldwide, especially sports people who suffer from trauma and knee injury [1,2]. Because of its lack of vasculature, lymphatic cells, and nerves, the self-healing ability of articular cartilage is limited. In addition, articular cartilage damage is a common cause of advanced osteoarthritis (OA), which shows severe pain, immobility, and dysfunction in the joint [1,3]. Current treatment to repair cartilage defects includes microfracture, autologous chondrocyte implantation, osteochondral autografts, and allograft-directed cartilage regeneration [4]. Moreover, the standard treatment strategies exhibit limitations and drawbacks, including fibrocartilage regeneration, donor site morbidity, inferior mechanical properties to hyaline articular cartilage, etc. Therefore, the regeneration of the cartilage defect is still challenging in today’s world [3,4,5].

Tissue engineering (TE) is a multifaceted field that combines material science and biology to replace or regenerate biological tissues [5]. Cell source, scaffold, and signaling molecules are the fundamental element for TE. The three-dimensional (3D) porous scaffold serves as a template for tissue formation by providing a suitable environment for tissues or organs to grow [5,6]. Among them, hydrogels promise 3D polymeric networks with a high water and porosity content for nutrient and oxygen diffusion. However, hyaluronic acid (HA), gelatin, etc., natural hydrogels exhibit the composition and the structure of the native extracellular matrix, which shows a distinctive ability for regenerative and tissue engineering applications. It can be extensively used for 3D cell encapsulation and surface seeding to develop a biomimetic construct for cartilage tissue engineering [7]. Regarding the cell source, the adipose-derived stem cells (ADSCs) have become the alternative source to bone marrow-derived mesenchymal stem cells (BM-MSCs). Compared to the BM-MSCs, ADSCs can proliferate rapidly and have a lower harvesting risk. ADSCs can be easily isolated and possess multilineage differentiation [8].

The HA is an anionic biopolymer made up of D-N-acetylglucosamine and D-glucuronic acid repeating units and serves as a backbone to form proteoglycan complexes. HA is renowned as the main extracellular matrix (ECM) component in all connective tissues. HA is also predominantly found in cartilage tissue [9]. A previous study stated that the cell surface receptors such as the cell adhesion molecule (CD44), named the homing cell adhesion molecule (HCAM), interact with HA and mediate the HA-receptor-mediated motility (HRAMM), therefore enhancing cell aggregation and instigating chondrogenesis and hyaline cartilage [10,11]. HA-based hydrogels for cartilage regeneration are becoming more common since they have demonstrated excellent biocompatibility and biological activity [11]. However, some clinical problems include the HA hydrogel’s lack of good mechanical properties, 3D structure, and unsatisfactory degradation rate. Fortunately, the carboxyl and hydroxyl groups of HA can also be chemically modified by methacrylate to yield the formation of crosslinked HA methacryloyl (HAMA) hydrogels upon UV light irradiation, which improves their chemical and mechanical properties, providing larger rigidities, high levels of viscoelasticity after swelling resistance to enzymatic degradation compared with unmodified HA, and biocompatibility preservation [11,12,13,14]. In addition, the applicability of HA-based hydrogels is still limited by their poor cell adhesiveness, which hinders cell proliferation [15]. Therefore, a potential strategy has been applied to overcome this challenge. The HAMA hybrid hydrogel has been formed by blending with other natural hydrogels enriched with ECM components, such as a gelatin-based hydrogel of gelatin methacryloyl (GelMA).

Gelatin is a natural hydrophilic polymer that hydrolyzes and denatures collagen with a lower immunogenicity because of the lower aromatic groups [16]. Gelatin contains various bioactive amino acid motifs, including arginine–glycine–aspartic acid (RGD) sequences that support cell adhesion and the progress of different cell types, matrix metalloproteinase (MMP) sequences, cell remodeling, etc. [17,18]. However, gelatin’s thermostability and mechanical modulus are poor when temperatures exceed 37 °C. Therefore, gelatin can also be chemically modified; the amino groups and minor hydroxyl groups presenting on the side chains of gelatin are replaced by methacryloyl groups in methacrylic anhydride crosslinked by additive chain reaction exposure to UV light and forms the gelatin–methacryloyl (GelMA) hydrogel. Because the amount of the free amino groups of gelatin was just about 0.3184 mmole per gram [17], meaning that even the amine group and hydroxyl group were replaced nearly wholly, the 100% degree of substitution (DS%) of the GelMA should contain less than 5.0 wt% of methacryloyl. This is why the GelMA structure also includes RGD sequences and MMP sequences of gelatin, which are not significantly affected. In that way, the cell adhesion property of the gelatin is retained in the GelMA hydrogel. Furthermore, GelMA hydrogels possess the better mechanical and tunable properties of most tissue engineering (TE) applications [12]. Nevertheless, GelMA is still not suitable for clinical applications alone because of its fast degradability rate, which may be based on internal metalloproteinase and a low degree of crosslinking by the low percentage of the free amino groups within the gelatin [12].

Some of the challenges in using the biological hydrogel for cartilage tissue engineering include that it will fail to mimic the mechanical and viscoelastic behavior of the native tissue and its excessive degradation rate. However, the development of a hybrid HAMA-GelMA (HG) hydrogel that incorporates several crosslinkers has emerged as a promising strategy for optimizing the physical and mechanical properties of the hydrogel so to make it suitable for cartilage regeneration or other biological applications [19,20]. For example, Shi et al. [21] showed this HG hydrogel through a dynamic covalent bond between phenylboronic acid-grafted hyaluronic acid (HA-PBA) and poly(vinyl alcohol), which was further stabilized through a secondary crosslinking between the acrylate moiety on the HA-PBA and the free thiol group from the thiolated gelatin with the antireactive oxygen species’ potential ability to enhance cartilage tissue regeneration. In addition, interest in free radical photopolymerization with a photoinitiator (P.I.) for crosslinking hybrid hydrogels with improved and regulating physiological/physicochemical properties has increased significantly in recent years [22]. Furthermore, lithium phenyl-2,4,6-trimethyl-benzoyl phosphinate (LAP) has been considered a promising P.I. that can be crosslinked via ultraviolet light and the blue light wavelength. The LAP has also shown promising results on cell viability in vitro [23].

In short, the hybrid hydrogels of HAMA and GelMA are considered promising biomimetic material, which requires a P.I. and light for the in situ crosslinking reaction [24]. For example, Hjortnaes et al. [25] have reported that the HG hydrogel shows valvular interstitial cell phenotype differentiation. In another work, Camci-Unal et al. [24] studied the HG hybrid hydrogel and showed increased hydrogel stiffness with a variety of different polymer concentrations. In addition, Teong et al. [26] stated that the stiffness of the HG hybrid hydrogel could induce the chondrogenesis activity on hADSCs.

Despite all previous studies, the HG hybrid hydrogel’s environmental, mechanical, and biological properties have not yet been completely established for the clinical use of chondral defect regeneration. Recently, nano-silica (nSi) has demonstrated biosafety combined with proper cell labeling and visualization in histological sections [27]. Some in the literature have also reported that nSi could enhance cell growth through ERK1/2 activation, improve osteoblast function, inhibit osteoclast function, and promote bone mineralization [28]. Bunpetch et al. [29] further reported that silicate-based bioceramic scaffolds could promote the osteogenesis of bone marrow stem cells (BMSCs) and contribute to maintaining a chondrocyte phenotype.

To the best of our knowledge, incorporating the acrylate-functionalized nSi nanoparticles into the HG hybrid hydrogels would be a promising solution for developing a 3D biomimetic scaffold for cartilage tissue engineering, which has not been reported thus far. The hypothesis is that the HG hybrid hydrogel incorporated with 3-acryloxypropyl silanetriol (APS)-functionalized nSi as a novel crosslinker (named for “acrylate functionalized nSi”; AFnSi) would be a better choice for tuning the physical, mechanical, and cellular activity and the increased chondrogenesis gene expression suitable for cartilage tissue engineering because the hydroxyl groups of the nSi surface can be grafted with multiple acrylates, which promotes the network crosslinking to form the C-C bond by the addition reaction within the HG hybrid hydrogels.

This study aimed to build a novel photocured hybrid hydrogel system comprising of HAMA, GelMA, and a minor AFnSi crosslinker to evaluate the cell viability and chondrogenic differentiation ability of human adipose-derived stromal cells (hADSCs). However, the lower cell toxicity of lithium phenyl-2,4,6-trimethyl-benzoyl phosphinate (LAP) is used as a photoinitiator. Figure 1 shows the overall concept of this study, i.e., the hADSCs loaded with photocrosslinked hybrid hydrogels to depict the preparations, physical/chemistry properties, and chondrogenic differentiation evaluation. Figure 1a–c shows the synthesis of the 2% (*w*/*v*) HAMA hydrogel, the 20% (*w*/*v*) GelMA hydrogel, and the hybrid hydrogels of the HG -AFnSi. The hybrid hydrogels of the HG -AFnSi system were photocured using LAP at 365 nm of UV light from Figure 1d. The physicochemical characteristics of these hybrid hydrogels were exhibited, including the swelling behavior, morphological conformation, mechanical properties, and biodegradation of the properties of the hybrid hydrogels. To further investigate cell viability and chondrogenic differentiation, the hADSCs were employed as cells loaded with the 2% HAMA–20% GelMA hybrid hydrogel with 0–1.0 (*w*/*v*) AFnSi crosslinker to examine the process of optimal chondrogenic development (Figure 1e), including in vitro cytotoxicity, chondrogenic differentiation gene expression, and sGAG, and the type II collagen results will explain whether or not this novel hybrid hydrogel design has excellent potential for cartilage tissue engineering applications in the future.

## 2. Materials and Methods

### 2.1. Materials

Hyaluronic acid (molecular weight of 2000 kDa) was purchased from Kikkoman (FCH-200, Tokyo, Japan). Gelatin from porcine skin (type B) and methacrylate anhydride (MA) (molecular weight of 154.16 Da) were purchased from Sigma-Aldrich, St. Louis, MO, USA). The P.I. of lithium phenyl-2,4,6-trimethyl-benzoyl phosphinate (LAP), nano-silica (SiO_2_; nSi), 3-acryloxypropyl trimethoxysilane (APMS) were also obtained from Sigma-Aldrich (St. Louis, MO, USA), and the sodium carbonate anhydrous (Na_2_CO_3_) and sodium bicarbonate (NaHCO_3_) were purchased from J.K Baker (Phillipsburg, NJ, USA). The other reagents, such as phosphate-buffered saline (PBS), Dulbecco’s Modified Eagle’s Medium (DMEM), fetal bovine serum (FBS), penicillin, and streptomycin were purchased from Gibco BRL (Grandland, New York, NY, USA). All other solvents were purchased from Merck (Darmstadt, Germany), TEDIA (Fairfield, CT, USA), or J. T. Baker (Phillipsburg, NJ, USA). These chemicals were all analytical/reagent grade and were used without further purification.

### 2.2. Synthesis of HAMA Hydrogel

The hyaluronic acid methacryloyl (HAMA) was produced using the previously described method [14,26]. Briefly, a 1 wt% HA solution of 100 mL was prepared from a 2/1 ratio of deionized distilled water (DDW) and dimethylformamide (DMF), which was stirred until the solution was used, and dissolved at 37 °C. Then, the 8 mL of methacrylic anhydride (MAA) was added to the HA solution and maintained a pH of 8–9 with 3N of sodium hydroxide and stirred continuously for 24 h at 4 °C. The methacrylate-modified HA solution was dialyzed against DDW for three days to remove the unreacted methacrylate group for the purification process. The HAMA product was lyophilized and stored at 4 °C for further studies.

### 2.3. Synthesis of GelMA Hydrogel

The gelatin methacryloyl (GelMA) was also produced using the previously described method [14,17] and modified as follows: the 10 g of gelatin was dissolved in 100 mL of 0.25 M buffer solution (10 *w*/*v*%) containing carbonate–bicarbonate (NaHCO_3_ and Na_2_CO_3_) until the clear solution occurred at 50 °C. The initial pH adjustment at pH 9 by 5 M sodium hydroxide or 6 M hydrochloric acid. Subsequently, 1 mL of liquid MAA was added to a 10 *w*/*v*% gelatin solution with a gelatin weight of 10 g under magnetic stirring at 50 °C under 500 rpm. The reaction proceeded for 3 h, and then the pH was readjusted to 7.4 to stop the reaction. After being filtered, dialyzed against the double-distilled water for 3 days, and lyophilized, the samples were stored at −80 °C until further use.

### 2.4. Synthesis of Acrylate Functionalized nSi Crosslinker (AFnSi)

To prevent nanoparticle aggregation, enhance dispersion stability, and act as a photocured coupling agent in an HG hybrid hydrogel network, this study used 3-acryloxypropyl silanetriol (APS)-functionalized nSi as reinforcing particles. Firstly, the 0.5% (*w*/*v*) of 3-acryloxypropyl trimethoxysilane (APMS) solution was prepared in 100 mL of 95% ethanol and stirred at 37 °C. The hydrolysis reaction changed the APMS into APS at pH 4 for 48 h. Finally, the pH was adjusted to 7 and the structure of APS was obtained and identified using Fourier transform infrared (FTIR) spectrometer. Subsequently, 10 g of nSi powder was added to the 100 mL of APS solution and stirred for 24 h to form the colloid solution ultimately. Then, the APS/nSi colloid solution was stirred in an oil bath at 100 °C for 4 h to carry out the condensation reaction between hydroxyl groups. Then, the 50 mL ethanol was added and centrifuged for 10 min at 8000 rpm. The sediment after centrifugation was collected and dried. The functional structure of acrylate-functionalized nSi (AFnSi) obtained was also identified by FTIR analysis.

### 2.5. Identification of the Synthesis of HAMA and GelMA

The proton NMR spectroscopy (Varian Gemini-200, Morgantown, PA, USA) was recorded for HA, HAMA, gelatin, and GelMA hydrogels to determine the incorporation of the methacrylate group. A total of 20 mg of each sample is completely dissolved in 1ml of deuterium oxide (D_2_O)-containing 3-(trimethylsilyl) propionic-2,2,3,3-d_4_ acid sodium (TMSP) salt serves as an internal standard. The degree of methacrylation (DoM) for HAMA was calculated by formulation (1). A brief explanation is as follows to determine the degree of methacrylation of HAMA is by ^1^H NMR analysis. The intensity of the signals of the vinyl groups (signals can typically be found around ~5.8 and 6.2 ppm) of the methacrylates are quantified and compared with a known peak in the spectrum. Sometimes the DoM is also quantified by quantifying the methacrylate methyl peak (at ~1.9 ppm) [14,29]. This study method looked at the sum of the methacrylate protons at δ 6.2 ppm and δ 5.8 ppm concerning the three protons on the methyl groups of the N-acetyl-glucosamine subunit at δ 2.0–2.1 ppm. The DoM calculation formula is shown in Equation (1), where the numerator and denominator are compared with the integral value of each proton. In addition, the number of free amine groups and methacryloyl groups in GelMA was also quantified by ^1^H-NMR assays and cited to the DoM (%) by the literature [17]. The quantification of both methacrylamide and methacrylate groups in GelMA was done simultaneously. However, the DoM (%) of target 90–100% checked the signal of the free amino groups disappeared in gelatin (signals can typically be found around ~3.0 ppm).
(1)$$\mathrm{DoM}\%\;=[\left(\mathrm{vinyl}\;\mathrm{groups})/2]/[\left(\mathrm{methacrylate}\;\mathrm{methyl}\;\mathrm{peak}\right)/3\right)]\times 100\%$$

### 2.6. Identification of the Synthesis AFnSi Crosslinker

The infrared spectra (IR) of the nSi_,_ 3-acryloxypropyl trimethoxysilane (APMS), and acrylate-functionalized nSi (AFnSi) were recorded for functional groups of the chemical structure using FTIR spectroscopy (system 2000 FT-IR, Perkin Elmer, Waltham, MA, USA) in the attenuated reflection (ATR) mode. The transmittance readings of the samples were measured by accumulating 32 scans at a resolution of 4 cm^−1^ in the spectral region of 4000–400 cm^−1^. The X-ray powder diffraction (XRD) was used for phase identification of crystalline material for the nSi and AFnSi samples and performed on a Bruker D8 advance diffractometer (Germany). The following parameters were set for the apparatus: Cu-K radiation using a graphite monochromator, a voltage of 40 kV, an electric current of 40 mA, and a scan range from 10° to 60° at a scanning speed of 2°/min. Transmission electron microscopy (TEM) was also used to examine the size and morphology of these nanoparticles. That means that the nSi and AFnSi suspension samples were dropped onto a copper grid coated with a carbon/formvar support film using a Pasteur pipette for TEM observation. After 15 s, the excess sample was blotted with filter paper and dried at room temperature. The grid was placed in a specimen holder and inserted into a 200 kV Joel JEM-2100 TEM for observation.

### 2.7. Fabrication of HG Hybrid Hydrogels

The HAMA hydrogel solutions were prepared in the concentration of 1% (*w*/*v*) and GelMA in the concentration of 10% (*w*/*v*), respectively. Then, the hybrid hydrogel by 2:1 volume ratio of HG solution was mixed with 0.3% (*w*/*v*) LAP P.I. at different concentrations of AFnSi crosslinker 0.1, 0.5, and 1% (*w*/*v*), respectively. The hybrid hydrogel made without AFnSi crosslinker was also as a control group. The hybrid hydrogel solutions were vortexed for homogenous mixing and photocured under UV light using the UV curing chamber from XYZ printing (USA) at a 280–400 nm wavelength for 120 s. However, these hybrid hydrogels were denoted as HAMA-GelMA (HG), HAMA-GelMA + 0.1% (*w*/*v*) AFnSi (HG+0.1% AFnSi), HAMA-GelMA + 0.5% (*w*/*v*) AFnSi (HG+0.5% AFnSi), and HAMA-GelMA + 1% (*w*/*v*) AFnSi (HG+1% AFnSi).

### 2.8. Characteristics of HG Hybrid Hydrogels

#### 2.8.1. Swelling Ratio Evaluation

To prepare the hybrid hydrogel samples for the swelling ratio measurement, a 300 µL of the hybrid hydrogel by 2:1 volume ratio of HG solution, each of different concentrations of AFnSi crosslinker, such as 0, 0.1, 0.5 and 1.0% (*w*/*v*) with 0.3% (*w*/*v*) P.I. of LAP, was added to the cylindrical plastic mold. These hybrid hydrogel solutions of each group were then exposed to UV light at 280–400 nm for 120 s. After the photopolymerization procedure, each hybrid hydrogel sample containing different concentrations of AFnSi crosslinker was placed in an Eppendorf tube with 2 mL of PBS for 24 h to maintain the equilibrium. After 24 h, the swollen hydrogels were weighed, and gently removed the excess water by blotting with kimwipes. Four replicates were used for each hydrogel sample. The swelling ratio of the hydrogel sample was calculated using the following formula:Swelling ratio = [(W_w_ − W_0_)/W_0_](2)

W_w_: wet weight of the hydrogel sample;

W_0_: dry weight of the hydrogel sample.

#### 2.8.2. The Microstructure Morphology Analysis

The morphological characteristics of these hybrid hydrogels were observed after photocuring with the HG hybrid hydrogel with different concentrations of AFnSi crosslinkers of 0, 0.1, 0.5, and 1.0% (*w*/*v*). However, these hybrid hydrogels were prepared as previously described for the swelling ratio samples and the cross-section area obtained after the freeze-dried method. Micrographs or the element mapping analysis of all the hydrogel samples were taken using a scanning electron microscope (SEM, JEOL, Tokyo, Japan) after these samples were coated with gold using a sputter coater under an ambient temperature. The average diameters of pores in HG hybrid hydrogel with different concentrations of 0, 0.1, 0.5, and 1.0% (*w*/*v*) AFnSi crosslinkers were evaluated using Image-J software (Media Cybernetics Inc., Rockville, MD, USA).

#### 2.8.3. Mechanical Properties Evaluation

These HG hybrid hydrogels with different concentrations of AFnSi crosslinkers of 0, 0.1, 0.5, and 1.0% (*w*/*v*) after photocuring were also measured for their compression strength and rheological behavior. The universal mechanical compression testing machine (Instron 5567, Noorwood, MA, USA) was used for compressive mechanical testing. The speed of the crosshead was 4 mm/s, and the loading cell was 200 N. The compression strength was calculated as stress when the compressive strain in the hydrogel reached 60%. The compression test was repeated for five samples of each hybrid hydrogel. In addition, the storage modulus (G′) and loss modulus (G″), and loss factor (Tan δ), as a function of the angular frequency (rad/s) behavior of these hybrid hydrogels was tested using HR-2 Discovery Hybrid Rheometer-2 (TA Instruments, New Castle, DE, USA) with the attachment of the 20 mm parallel plate. Each hybrid hydrogel was added to the 0.5 mm gap between the plates and waited until the normal force became zero. Firstly, frequency sweeps (ω; rad/s) were tested at a constant 1% strain at 37 °C to check the elasticity properties of the hybrid hydrogel. Furthermore, the storage modulus (G′) as a function of the temperature and time sweep was also carried out by increasing the temperature with 0.5 °C/min heating rate from 10 to 60 °C at a frequency of 1 rad/s.

#### 2.8.4. In Vitro Degradation Assay by Hyaluronidase

As mentioned previously in the swelling ratio experiment, these hybrid hydrogels were prepared and allowed to swell in PBS for 24 h to reach the swelling equilibrium first. Then, these hybrid hydrogels were incubated in 1 mL of PBS buffer containing 2.6 Uml^−1^ hyaluronidase (human plasma physiological concentration) at 37 °C. These hybrid hydrogels were collected every 5 days and weighed after being blotted for 30 days. However, the fresh hyaluronidase solution was replaced every day, and four replicates were used for each hybrid hydrogel. The degree of degradation of the hybrid hydrogels was calculated by normalizing the residual hydrogel wet weight and initial hydrogel wet weight.

### 2.9. Cell Viability and Chondrogenesis Assay of HG Hybrid Hydrogels

#### 2.9.1. Isolation and Culturing of hADSCs

This study examined the isolation of hADSCs from human subcutaneous adipose tissue, which was done according to a previously described procedure [11,26]; the ADSCs were isolated from subcutaneous adipose tissue obtained from human patients during orthopedic surgery after obtaining informed consent from all the patients and approval from the Kaohsiung Medical University hospital ethics committee (KMUH-IRB-E(II)-20150193). Briefly, 3 g of human subcutaneous adipose tissue was extracted and cut into small pieces using scissors. The minced tissues were digested with 1 mg/mL of type Ⅰ collagenase at 37 °C under 5% CO_2_ for 24 h. Then, centrifugation was performed at 1000 rpm for 5 min. The pellet was collected and washed with PBS twice. After that, the pellet was resuspended in a K-NAC medium, and the cells were counted and plated in a 100 mm culture dish. Subsequently, the ADSCs attached to the culture plate were maintained at 37 °C under a 5% CO_2_ incubator. The K-NAC medium used in this study is suitable for the isolation and expansion of ADSCs described in the initial study; the K-NAC medium mainly contains Keratinocytes-SFM (Gibco BRL, Rockville, MD, USA) supplemented with 25 mg of bovine pituitary extract (BPE), 2.5 µg of human recombinant epidermal growth factor, 2 mM N-acetyl-l-cysteine, 0.2 mM L-ascorbic acid, and 5% FBS. The first medium change was performed after 24 h, and the unadhered ADSCs to the plate were washed off using PBS. Subsequently, the fresh medium was changed every two days. The cells could grow nearly 90% confluence and subculture for further cell study experiments.

#### 2.9.2. Cell Viability Assay

The HG hybrid hydrogel with different concentrations of 0%, 0.5%, and 1.0% (*w*/*v*) of AFnSi crosslinker were prepared according to the protocol mentioned above. Briefly, 200 µL of each hydrogel sample was coated on the 24-well plates. Each hydrogel sample was crosslinked using UV light at 365 nm. To each crosslinked hydrogel sample, 1ml of PBS was added and incubated for 24 h at 37 °C under 5% CO_2_ to reach the swelling equilibrium rate. Afterwards, the PBS was removed gently. The samples can be further used to perform cell viability and differentiation analysis.

The CellTiter96^®^ aqueous one-solution cell proliferation assay (MTS) kit was used to determine cell viability (Promega, Madison, WI, USA). In other words, the HG hybrid hydrogel with the different concentrations of 0%, 0.5%, and 1.0% (*w*/*v*) AFnSi crosslinkers and the HA hydrogel (control group) in hADSCs were tested by MTS assay; all the sample groups were cultured in DMEM medium with 1 × 10^6^ cells for day 1, 3, and 5 at 37 °C under 5% CO_2_. Briefly, the culture media after incubation in each sample well at each time-indicated point was removed and replaced by 200 µL of fresh medium and 40 µL of MTS reagent to each sample well in a 24-well culture plate. The plates were incubated for 4 h at 37 °C under 5% CO_2_. Finally, 100 µL of the solution was pipetted out from each sample well plate and placed in a 96-well dish. The optical density (OD) at 490 nm was taken using a 96-well ELISA plate reader (Synergy H1, BioTek, Winooski, VT, USA).

Furthermore, the cell toxicity evaluation for the HG hybrid hydrogel with the different concentrations of 0%, 0.5%, and 1.0% (*w*/*v*) of AFnSi crosslinkers and HA hydrogel in hADSCs were evaluated using live/dead staining for day 1 and day 5. The live/dead staining assay was performed according to the manufacturer’s protocol. Briefly, 20 µL of ethidium homodimer-1 (EthD-1) and 5µL calcein-AM were mixed with 10 mL of PBS solution and vortexed. A total of 500 µL of working dye solution was added to each well containing the samples and incubated for 1 h at 37 °C under 5% CO_2_ conditions. Finally, the live and dead cells in the samples were observed using an inverted fluorescence microscope (Leica DMi8 Inverted Fluorescence Microscope).

#### 2.9.3. Chondrogenic Marker Gene Expression

The chondrogenic effect of the HG hybrid hydrogel with 0% and 0.5% (*w*/*v*) of AFnSi crosslinkers and HA hydrogel (control group) in hADSCs were examined for chondrogenic marker gene expression using quantitative real-time PCR assay. Each sample was cultured with 1 × 10^6^ cells in 24-well plates in basal medium maintained at 37 °C under 5% CO_2_ condition for days 1, 3, 5, and 7. At each time-indicated point, the HG hybrid hydrogel samples were collected. Total RNA extraction was performed using TRIzol reagent. The quality of the RNA was confirmed using a Thermo scientific NanoDrop^TM^ 1000 spectrophotometer (Thermo Fisher Scientific, Waltham, MA, USA). RNA to cDNA was reverse-transcribed using the TOOLs easy Fast RT kit (Taipei, Taiwan). Real-time PCR was performed using IQ SYBR green supermix (Bio-Rad Laboratories, Hercules, CA, USA). The transcribed cDNA samples were analyzed for the genes of interest comprising SOX-9, aggrecan, type Ⅱ collagen, and type Ⅰ collagen. The primers used in this study are listed in Table 1. After completing the real-time PCR, the dissociation curve was produced to check the specificity of each PCR product. All values of the gene of interest were normalized to the expression of glyceraldehyde-3–phosphate–dehydrogenase (GADPH) level with an average threshold cycle (Ct) value using the comparative method. The experiment was repeated three times at every time-indicated point.

#### 2.9.4. Quantification of DNA, sGAG Deposition, and Collagen Type Ⅱ Synthesis

The DNA content for each sample was measured using a Hoechst dye assay, and the calf thymus was used as a standard curve to measure the DNA content. The analysis of sulfated glycosaminoglycan’s (*sGAG*) assay protocol was performed with the Blyscan^TM^ kit (Biocolor Ltd., Carrickfergus, Northern Ireland), and the chondroitin solution with different concentrations from 0 to 25 µg/µL was used as the standard for dimethylmethylene blue (DMMB) assay. Finally, the optical density measurement at 650 nm was taken using an ELISA plate reader (Synergy H1, Biotek, Winooski, VT, USA). The DMMB assay was performed to detect and quantify the amount of sGAG content in the HG hybrid hydrogel with 0% and 0.5% (*w*/*v*) of AFnSi crosslinkers and HA hydrogel in hADSC. These samples were cultured with 1 × 10^6^ cells in 24-well plates for days 5 and 7 in a basal medium maintained at 37 °C under 5% CO_2_. These samples were harvested, washed with PBS, and digested at each indicated point sing a Papain solution for 15 h at 60 °C. Enzyme-linked immunosorbent assay (ELISA) was also used to quantify collagen type Ⅱ present in the HG hybrid hydrogel with 0% and 0.5% (*w*/*v*) of AFnSi crosslinkers and HA hydrogel in hADSCs. Each sample was cultured in a basal medium with 1 × 10^6^ cells in 24-well plates for days 5 and 7, respectively. At each time-indicated point, these samples were harvested, and the type Ⅱ collagen content in each sample was measured using a type Ⅱ collagen detection kit (Chondrex, Redmond, WA, USA).

### 2.10. Statistical Analysis

The scoring data were statistically analyzed to express the mean ± SD (*n* = 3~6). A one-way ANOVA (or *t*-test method) was performed, and the * *p* < 0.05 and ** *p* < 0.001 were compared to the control or compared between treatment groups.

## 3. Results and Discussion

### 3.1. Identification of the Synthesis HAMA and GelMA

The methacrylation process involved adding a methacryloyl group to the amine and hydroxyl residues of gelatin and HA (see Scheme Figure 2) [17,24,26]. The well biotolerance hyaluronic acid methacryloyl (HAMA) is gaining popularity due to its ability to gelatinize via simple photocrosslinking. In particular, the biocompatibility of HAMA after photocuring showed no cytotoxicity. Additionally, seed cells can be encapsulated in HAMA, allowing the cell to live in a 3D environment for a while [14,22]. Figure 2a represents the synthesis of the hyaluronic acid methacryloyl (HAMA), in which the primary hydroxyl group of hyaluronic acid reacts with the methacrylate pendant group of methacrylic anhydride (MA) to form the hyaluronic acid methacryloyl (HAMA) hydrogel. The methacrylamide-modified gelatin has natural and synthetic gelatin properties, including cell adhesion sites and tunable mechanical properties [17]. As illustrated in the schematic Figure 2b, the unsaturated bonds were grafted onto the gelatin via a simple one-step conversion of anhydride to an amino and hydroxyl group. Successfully, the gelatin methacryloyl (GelMA) hydrogel was obtained.

The degree of methacrylation (DoM) is an essential characteristic of the crosslinking density of the hydrogel matrix that has a significant impact on the mechanical properties, structure, porosity, swelling, and degrading abilities [24,30]. Figure 3 illustrates the methacrylation of both biopolymers (HAMA, GelMA) which was verified by ^1^H NMR spectra. Figure 3a,c shows the ^1^H NMR spectra of hyaluronic acid and gelatin. Figure 3b is the ^1^H NMR spectra of HAMA, which shows methacryloyl peaks at 5.5 and 5.8 ppm. A peak at 2.16 ppm corresponds to the grafted methacryloyl group’s methyl protons (CH_2_=CH(CH_3_)), independent of the xunaltered HA spectra. HAMA with an optimal DoM creates stiffer hydrogels, which are proven to promote cell adhesion in tissue engineering applications. Based on the NMR spectra, the calculated DoM of HAMA was about 85 ± 10%, respectively.

Figure 3d also shows the ^1^H NMR spectra of GelMA, which shows methacryloyl peaks at ~5.5 and ~5.8 ppm, respectively, which coincide with the acrylate protons (CH_2_=CH(CH_3_)) of hydroxylysine and lysine. Although both methacrylamide and methacrylate groups are present in GelMA, the content of methacrylate is much lower than that of methacrylamide. However, with a peak at 1.92 ppm, this increase corresponds to the grafted methacryloyl group’s methyl protons (CH_2_=CH(CH_3_)) compared with the unaltered gelation spectra. Therefore, the quantification of the methacrylamide groups in GelMA, and the DoM (%) of the target 90–100%, has confirmed that the free amino groups disappeared in the gelatin (signals can typically be found around ~3.0 ppm).

### 3.2. Identification of the AFnSi Crosslinker

Due to nanoparticles’ small size and very high surface energy (NPs), NPs are prone to aggregation in solvents or other liquid matrices. Some experiences prevent nSi agglomeration and increase NPs surface reactivity due to surface modification by physical and chemical treatments [31,32]. Furthermore, one of the most common methods to modify the nSi surfaces is an organo-silane chemical treatment, which can establish a strong chemical bond between the NPs surfaces and polymer chains [28,33]. In addition, nSi have demonstrated biosafety and can enhance cell growth [27,28,34].

In this study, we prepared a novel acrylate-functionalized nSi (AFnSi) crosslinker to stabilize the 3D network of the photocrosslinkable HG hybrid hydrogel, of which the schematic diagram of the surface modification steps is shown in Figure 4a. The FTIR spectra results of Figure 4b expressed that the AFnSi bands have the new band at 1869 cm^−1^ and 1692 cm^−1^ for the C=O groups, and 1611 cm^−1^, which shows that the strong C=C stretching also appeared in the AFnSi, which was different from the spectra obtained from the nSi. The results of the FTIR spectra indicate that the AFnSi NPs have been successfully fabricated. The morphologies of the nSi and AFnSi NPs were studied using TEM and are shown in Figure 4c,d. Figure 4c,d shows the TEM images of the nSi and AFnSi NP crosslinker. The average size of the as-received NPs of nSi has approximately 16 nm (Appendix A). After the surface modification of the APS on nSi, the size of the AFnSi was also around 15 nm (Appendix A), which was not significantly different than the unmodified nSi. This indicated the AFnSi were wrapped in polymer chains, but the smaller molecule of APS grafted on nSi did not affect its size much after drying. The crystallinity of the nSi and AFnSi crosslinkers was analyzed by electron beam diffraction by TEM, showing that both are crystallinity structures of an amorphous nature, which is the same as the XRD result, as shown in Appendix A.

### 3.3. Swelling Ratio of HG Hybrid Hydrogels

Hydrogels contain over 90% water and can retain it within their three-dimensional crosslinked structures [35]. Hydrogels’ swelling ability is a measure of their hydrophilicity and is determined by the size of the hydrogel pores. This fundamental characteristic has been demonstrated to impact cellular activity [36]. To build the 3D network, hydrogels are crosslinked, and the degree of crosslinking influences the structure and swelling capacity of the hydrogels, as well as their integrity. Hydrogels are networks of crosslinked hydrophilic polymer networks, which, instead of dissolving in water, undergo swelling [24].

Figure 5 indicates the swelling behavior of the photocrosslinked HG hybrid hydrogel with the different concentrations of the AFnSi crosslinkers. The concentrations of the AFnSi crosslinkers incorporated into the HG hybrid hydrogels are 0, 0.1, 0.5, and 1% (*w*/*v*), respectively. It has been observed that the swelling behavior of the HG hybrid hydrogel with different concentrations of AFnSi crosslinker was found to be tunable. The swelling ratio of the HG hybrid hydrogel with 0.1% (*w*/*v*) AFnSi, for example, decreased from 60%. Similarly, the swelling ratio significantly reduced to about 35% in the HG hybrid hydrogel with 0.5% and 1% (*w*/*v*) AFnSi compared to the HG hydrogel alone (~74%). It was predicted that these results would occur, since an increased crosslinking concentration allows for greater crosslinking density, which has been previously reported [24]. Therefore, the HG hybrid hydrogels with higher AFnSi crosslinker concentrations should have smaller pore sizes and cause minor swelling when compared to the HG hybrid hydrogels made with lower crosslinking concentrations. These findings demonstrate that the swelling behavior of the HG hybrid hydrogels may be controlled via crosslinker concentrations. These different swelling ratios can be used as parameters for the future assessment of cell viability and chondrogenesis capacity.

### 3.4. Morphological Examination of HG Hybrid Hydrogels

The internal structure and morphology of the scaffolds are also essential for their intended application in TE because they determine how well the scaffolds perform in the microenvironment [37]. Figure 6 shows the microstructure of the HG hybrid hydrogel with different concentrations (0~1.0%) (*w*/*v*) of AFnSi crosslinkers from the SEM. Despite the possibility that lyophilization creates artificial pores, all of the HG hybrid hydrogels were made under identical experimental circumstances and lyophilized at the same temperature for the same amount of time so to eliminate any variations. All of the HG hybrid hydrogels have honeycomb-like structures from the linked interior photocured molecules (HAMA, GelMA) and the AFnSi crosslinker. However, these SEM images indicate the decrease in pore size when the concentration of the AFnSi crosslinker increases, which may be due to the increase in the degree of crosslinking in the HG hybrid hydrogels. The average diameters of the pores in the HG hybrid hydrogel with the different concentrations of 0, 0.1, 0.5, and 1.0% (*w*/*v*) AFnSi crosslinkers evaluated were 95 ± 0.8 µm, 86 ± 0.6 µm, 75 ± 0.2 µm, and 76 ± 0.2 µm, respectively. The average pore size has no significant difference between the HG+0.5% (*w*/*v*) AFnSi and the HG+1.0% (*w*/*v*) AFnSi. This is probably due to the addition of AFnSi exceeding 1.0% (*w*/*v*), which will cause the difficult problem of further uniform dispersion in the HG hybrid hydrogel system, as indicated from our observations and also from Appendix A, which also showed a similar silicon element dispersion density. The HG hybrid hydrogels’ pore connectivity ensures that they can conduct nutrient exchange and cell migration, indicating that they have the potential to be applied in biomedical applications. The photocuring of the HG hybrid hydrogel using AFnSi crosslinkers facilitates these different degrees of porous network structure formation. In addition, the higher crosslinking and homogeneous pores should improve the mechanical properties and suppress the fast degradation problem of the biohydrogel and make it easier to use in tissue engineering applications.

### 3.5. Mechanical Properties of HG Hybrid Hydrogels

In this work, a universal testing machine (Noorwood, MA, USA) was used to test the mechanical properties of HG hybrid hydrogels under a compression model to determine the relationship between the amount of AFnSi crosslinker added and their compression properties. Figure 7a shows the schematic photos of the typical compression process for photocrosslinked HG hybrid hydrogels with the different concentrations (0~1.0 *w*/*v*) of AFnSi crosslinkers. Generally, the conventional hydrogels derived from natural materials are typically weak mechanically and have limited tissue engineering and biomedical applications. There is a great demand for excellent hydrogel scaffolds with desirable mechanical compressive stability and recovery. However, Figure 7b indicated that HG hybrid hydrogels incorporated with the 0.5 and 1.0 (*w*/*v*) AFnSi crosslinkers (HG+0.5% AFnSi; HG+1.0% AFnSi) can be resistant to more deformation (~50% strain) and have the ability to retain up to 85% of their original dimensions after the compression process. The HG hybrid hydrogels alone showed lower levels of stress (102 kPa) and deformation (35% strain), and the maximum compressive stress in the hybrid hydrogel of HG+0.5% AFnSi increases by nearly two times of the HG alone.

In addition, the compressive modulus also increased by about 35 ± 1.5 kPa from the HG hybrid hydrogels incorporated with the 0.5% (*w*/*v*) AFnSi crosslinkers (HG+0.5% AFnSi), which are approximately three times higher than the compressive modulus of the HG hybrid hydrogel alone (Figure 7c). There is no significant difference in the modulus between the HG+1.0% AFnSi and the HG+0.5% AFnSi, but the compressive stress stability of the HG+0.5% AFnSi is still more than the HG+1.0% AFnSi crosslinker formulations (Figure 7b). It is speculated that the addition of AFnSi exceeding 1.0% (*w*/*v*) will cause the problem of the uneven dispersion of AFnSi in the HG hybrid hydrogel system.

The experiments carried out under the frequency dependence of the viscoelastic properties of all the photocrosslinked HG hybrid hydrogels in the linear (at 1% strain) regions were also analyzed. This means the storage modulus, loss modulus, and loss factor are approximately constant in the linear area. Figure 8 shows the effect of frequency (ω) on the dynamic properties at a constant strain amplitude, and the sweep frequency is from 1 to 100 Hz, and the storage modulus G’ and the loss modulus G’’ showed stable levels in the region of low to the high angular frequency in Figure 8a. However, these results also showed that the storage modulus is always higher than the loss modulus. Obviously, the loss factor is always less than 1, so all the photocrosslinked HG hybrid hydrogels have a predominantly elastic response within frequencies from 1.0 to 100 Hz. In particular, the G’ of the photocrosslinked hybrid hydrogels of HG+0.5% AFnSi and HG+1.0% AFnSi also have an excellent range which is greater than G”, as seen in Figure 8a,b, which indicate that both hybrid hydrogels have high elasticity compared with the HG hybrid hydrogel alone. In other words, the loss factor (Tan δ) of both photocrosslinked hybrid hydrogels of HG+0.5% AFnSi and HG+1.0% AFnSi were lower than that of the other HG hydrogels during the 1.0–100 Hz frequencies, which exhibited a more dominant elastic behavior. However, these HG hybrid hydrogels have thermal stability across the monitored temperature range (Appendix A), with an oscillatory temperature sweep between 20 °C and 45 °C at a rate of 0.5 °C/min heating rate and a frequency of 1 rad/s.

The polymeric chains are tightly coupled together in highly crosslinked hydrogels via covalent crosslinks, which may prevent chain mobility. A lower crosslink density may facilitate movement between chains and a more relaxed behavior. Therefore, the appropriate amount of crosslinking agent can provide more viscoelastic properties or have a lower loss factor. This is the reason why, when extra force is removed from HG hybrid hydrogels with the 0.5% (*w*/*v*) AFnSi crosslinker, they should return to their original shape with a minimal energy loss compared with the 1.0% (*w*/*v*) AFnSi crosslinker (Figure 8). According to previous literature, the mechanical properties of the ECM have profound effects on the progression of various cellular functions, such as signal transduction, gene expression, proliferation, differentiation, and ECM secretion [38]. However, human bone and cartilage are often affected by compression behavior in daily life; as a result, the hybrid hydrogels which use HG+0.5% AFnSi and HG+1.0% AFnSi as tissue engineering scaffolds in vitro and in vivo could be highly beneficial.

### 3.6. Degradation Study of HG Hybrid Hydrogels

The biodegradability is essential for tissue engineering scaffolds because it allows them to be modified for the regeneration process or for the release of bioactive molecules that have been encapsulated within them [39]. In many cases, engineered hydrogel-based scaffolds are designed to degrade within the body following implantation at a rate comparable to the rate at which new tissue is formed. A slow degradation process may provide a stable culture environment, facilitating cell migration and differentiation activity, and tissue regeneration time. Hydrogel degradation can be triggered by adding enzymes or chemicals to the solution. For example, hyaluronidase degrades HA in a short period. Because of this low long-term viability, it is challenging to complete the long and complex regeneration process of cartilage [40].

The enzymatic degradation results of the HG hybrid hydrogel by hyaluronidase are shown in Figure 9, which can degrade the HAMA component of the HG hybrid hydrogels. However, these results were investigated to see how the AFnSi crosslinker of different concentrations affects the degradation rate. As expected, increasing the concentration of AFnSi crosslinker from 0 to 1.0% (*w*/*v*) in the HG hybrid hydrogel can slow down the degradation rate. After 30 days, the hybrid hydrogel of the HG alone was 58% entirely degraded under 2.5 U/mL hyaluronidase at 37 °C, whereas the hybrid hydrogel of HG+0.1% AFnSi and HG+0.5% AFnSi had about ~46% and ~25% degradation, respectively. Furthermore, the degradation rate of the hybrid hydrogel of HG+1.0% AFnSi under 2.5 U/mL hyaluronidase at 37 °C enzymatic degradation was quite close to that of the HG hybrid hydrogel alone without hyaluronidase.

However, the enzymatic degradation rate of the HG hybrid hydrogels and their swelling ratio, microstructure, and crosslinking degree have been demonstrated to be correlated. On the other hand, the degradation data findings confirmed the observation that the degradation rate is inversely proportional to the compression modulus of the HG hybrid hydrogel. Therefore, the higher crosslinking degree and compression modulus of the HG+0.5% AFnSi and HG+1.0% AFnSi hybrid hydrogels decrease as the degradation rate increases. However, the optimal HG-crosslinked hybrid hydrogel system with an AFnSi crosslinker should improve the stability so to protect the biohydrogel from rapid degradation, while also providing long-term mechanical support.

### 3.7. Cell Viability Evaluation of hADSCs on HG Hybrid Hydrogels

The cell viability was assessed in HA as a control, as well as the other samples comprising the photocrosslinked hybrid hydrogels of HG, HG+0.5% AFnSi, and HG+1.0% AFnSi during the culture period of 5 days, as shown in the Figure 10. All the groups were incubated at 37 °C with 5% CO_2_ in basal medium. The MTS assay evaluated the mitochondrial conversion of tetrazolium salt to water-soluble formazan crystals in the growing cells. Compared to the HA control culture method, the hADSCs cultured in the crosslinked HG hydrogels (HG alone, HG+0.5% AFnSi, and HG+1% AFnSi) did not affect cell viability; they still steadily maintained or increased compared to the control group from day 0 to day 5. In other words, no cell toxicity variation was identified between the HA and HG hybrid hydrogels.

Additionally, we observed biohydrogel in the basal medium by fluorescence imaging of the hADSCs laden in a three-dimensional environment, as shown in Figure 11. To determine whether these HG hydrogels can affect cell viability behavior, the live/dead assay, which comprises the HA, is the photocrosslinked hybrid hydrogels of HG, HG+0.5% AFnSi and HG+1.0% AFnSi.

Results in all groups indicated that most of the hADSCs were still alive on day 1 and day 5, with only a small number of dead cells seen, which comprised the HA and the photocrosslinked hybrid hydrogels of HG, HG+0.5% AFnSi, and HG+1.0% AFnSi. After 5 days of incubation, the number of dead cells found had minor increases in all groups. It is worth mentioning chondrogenesis proliferation stages where MSC-like cells (bone marrow or adipose tissue) should be induced to undergo aggregation and subsequent condensation [41]. Therefore, it is normal to have a spherical or aggregation morphology in the study. Taken together, hADSCs are not adversely affected by incubation with the HG hydrogel with the 0.5% and 1% (*w*/*v*) AFnSi crosslinker hydrogel scaffolds, and MTS has confirmed the favorable cytocompatibility required for further biological investigation. Among them, cell viability remained higher in the HG hydrogel with the 0.5% AFnSi crosslinker hydrogel. As a result, we found that the HG hydrogel with the 0.5% AFnSi crosslinker hydrogel scaffold has a low cytotoxicity profile and may be employed in vitro to assess chondrogenesis differentiation ability.

### 3.8. Chondrogenic Differentiation Ability of HG Hybrid Hydrogels

Cellular behaviors in tissue microenvironments are strongly affected by the ECM and soluble factors. In particular, the ECM is made of fibrous proteins such as collagen, fibronectin, and elastin, providing sufficient stiffness for the matrix in vivo within the range from 1 kPa to a few hundred kPa [42]. Previous studies have also reported that ADSCs would respond to the matrix stiffness because moderate stiffness (6–8 kPa) tended to induce better chondrogenic differentiation and caused the cells to synthesize a more hyaline-like cartilaginous matrix on day 5 [26]. In addition, bioinspired hydrogel structures have also been studied for tuning cell behavior and responses via mechanical property optimization or hydrogel stability enhancement [42]. However, nanoparticles can serve as crosslinkers, and, when added within hydrogels, can be used to modulate biomimetic 3D environments that reconstruct the complexity of native tissue for potential regenerative medicine [42].

The chondrogenic differentiation results in gene expression and sGAG formation in hADSCs for these HG hybrid hydrogels are shown in Figure 12 and Figure 13. The hADSCs seeded on the HA as control, the photocrosslinked hybrid hydrogels of HG, and HG+0.5% AFnSi were analyzed for chondrogenic differentiation at the time-indicated points on day 1, 3, 5, and 7 in the basal medium. In Figure 12, the relative expression of genes such as SOX-9, aggrecan, type Ⅱ collagen, and type Ⅰ collagen was determined using RT-PCR. In particular, we were looking for any upregulation of these genes at various time points. SOX-9 is a marker gene of chondrogenic differentiation at an early stage [26,40], which is significantly upregulated at day 5 on the photocrosslinked hybrid hydrogels of HG+0.5% AFnSi (Figure 12a). However, the expression of the SOX-9 gene of HG+0.5% AFnSi was increased by up to about 3.5 and 2.5-fold compared with the HA and HG alone on day 7, respectively. During the middle and late stages of the chondrogenic differentiation gene, the type II collagen gene is known to be involved [26,40]. Indeed, the upregulation of the type Ⅱ collagen gene was noticed in both photocrosslinked hybrid hydrogels of HG and HG+0.5% AFnSi compared with the control HA from day 3 to day 7, as shown in Figure 12c. Meanwhile, the aggrecan gene has also been identified as a late marker gene of chondrogenic differentiation. The experimental data of Figure 12b showed upregulation expression of aggrecan on both photocrosslinked hybrid hydrogels of HG and HG+0.5% AFnSi compared with the control HA from day 3 to day 7. In conclusion, the photocrosslinked hybrid hydrogels of HG+0.5% AFnSi showing the expression genes of SOX-9, collagen II, and aggrecan from day 3 to day 7 have the most significant increase when compared with the HA control group and HG alone. Conversely, the type I collagen gene expression is known as a fibrocartilage marker gene [43]. The expression of type I collagen was significantly downregulated from day 1 to day 7 in the basal medium when the hADSCs were cultured in both photocrosslinked hybrid hydrogels of HG and HG+0.5% AFnSi compared with the control HA.

Furthermore, the sGAG was found to be an essential ECM component in articular cartilage, and the amount of sGAG produced was found to be directly proportional to the level of chondrogenesis. Therefore, the DMMB assay was used to quantify the extracellular matrix production, indicated by the sGAG levels. It is noted that the total amount of sGAG and the average amount of sGAG/DNA were significantly higher on days 5 and 7 in the ADSCs cultured for both photocrosslinked hybrid hydrogels of HG and HG+0.5% AFnSi compared with the control HA, as shown in Figure 13a,b.

Collagen type II is also a significant component of articular cartilage. For articular cartilage to maintain a differentiated morphology and the associated secretion activities, type II collagen was expected to provide signaling molecules [26]. Unsurprisingly, it can be seen that both the total type II collagen and the collagen II/DNA content of the photocrosslinked hybrid hydrogel of HG+0.5% AFnSi show a significant increase compared with HG alone and HA, as shown in Figure 13c,d.

Based on the previous proliferation results by MTS, as well as the results of this chondrogenic differentiation genes study, and the chondrogenic secreted sGAG and type II collagen signaling molecules, it was determined that the photocrosslinked hybrid hydrogels of HG+0.5% AFnSi not only provided an environment for the proliferation of hADSCs when cultured in a basal medium, but also had the potential to support sGAG production and the secretion activities of type II collagen through the chondrogenic differentiation. In total, the addition of a 0.5% AFnSi crosslinker to the HG hybrid hydrogel scaffold, as demonstrated by these findings, could significantly enhance the chondrogenic differentiation of hADSCs.

## 4. Conclusions

There is a need to develop a bioinspired scaffold that can directly bond to tissue surfaces, which will be used in cartilage tissue engineering applications. Here, the photocurable HAMA-GelMA (HG) hybrid hydrogel with few acrylate-functionalized nano-silica (AFnSi) inorganic crosslinkers can exhibit variable pore size, mechanical properties, swelling ratio, and an acellular matrix that is highly biocompatible. The microstructure stability within the hybrid hydrogel was affected by the degree of crosslinking with the AFnSi amount. It was able to withstand additional force and large deformation, which allowed hybrid hydrogels to withstand additional pressure and large deformation so to recover to their original shape when the pressure was removed. In other words, although the concentration or degree of crosslinking from the AFnSi crosslinker can control the mechanical properties of the HG hybrid hydrogel, only the optimal crosslinked three-dimensional biohydrogel matrix may benefit from cell proliferation, migration, and morphogenesis in this study. However, we developed the optimal characteristics of pore size (75 ± 0.2 µm), swelling ratio (35%), and compression modulus (35 ± 1.5 kPa) for the novel photocrosslinked hybrid hydrogels of HAMA-GelMA (HG) + 0.5% (*w*/*v*) acrylate-functionalized nano-silica (AFnSi) crosslinker that allows the survival of human adipose-derived stem cells (hADSCs), and which can differentiate hADSCs into chondrogenesis. For example, the significant increase in the amount of sGAG deposition and the expression of chondrogenic marker genes (such as SOX-9, type II collagen gene, and aggrecan) within the photocrosslinked HG hybrid hydrogels with the 0.5% (*w*/*v*) AFnSi crosslinker compared with other groups. This bioinspired HG hybrid hydrogel with the AFnSi crosslinker would be a superior alternative for chondrogenesis and is a promising hydrogel for cartilage tissue regeneration studies in the future.

## Figures and Tables

**Figure 1 polymers-14-02003-f001:**
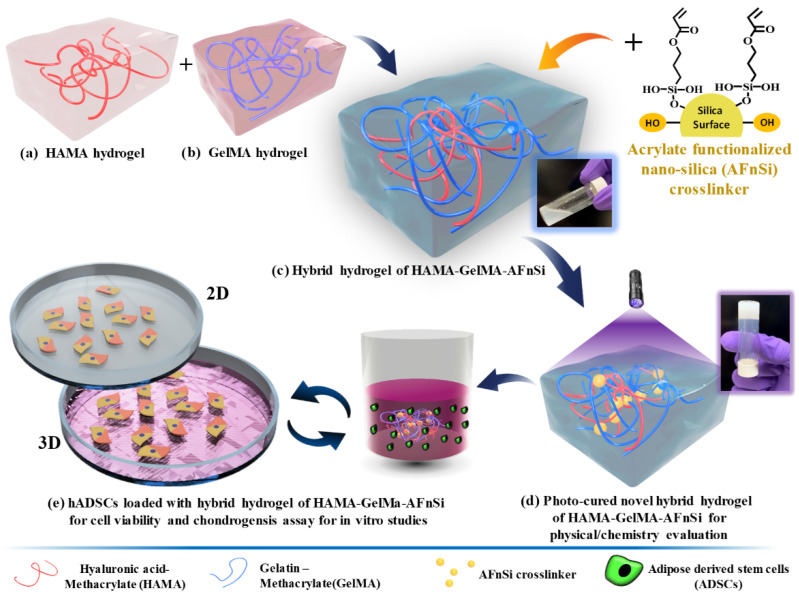
Schematic illustration of the experimental design and steps (**a**–**e**) for hADSCs loaded with optimal chondrogenic differentiation hybrid hydrogels comprised of hyaluronic acid methacryloyl (HAMA), gelatin methacryloyl (GelMA), and the acrylate-functionalized nano-silica (AFnSi) crosslinker in the in vitro studies. (**a**) 2% (*w*/*v*) HAMA hydrogel, (**b**) 20% (*w*/*v*) GelMA hydrogel, (**c**) the hybrid hydrogels of HG -AFnSi, (**d**) the hybrid hydrogels of HG -AFnSi system were photocured using LAP at 365 nm of UV light and their physicochemical characteristics were evaluated, and (**e**) the hADSCs loaded with the hybrid hydrogels of HG -AFnSi with 0–1.0 (*w*/*v*) AFnSi crosslinker to examine the process of cell viability and optimal chondrogenic development.

**Figure 2 polymers-14-02003-f002:**
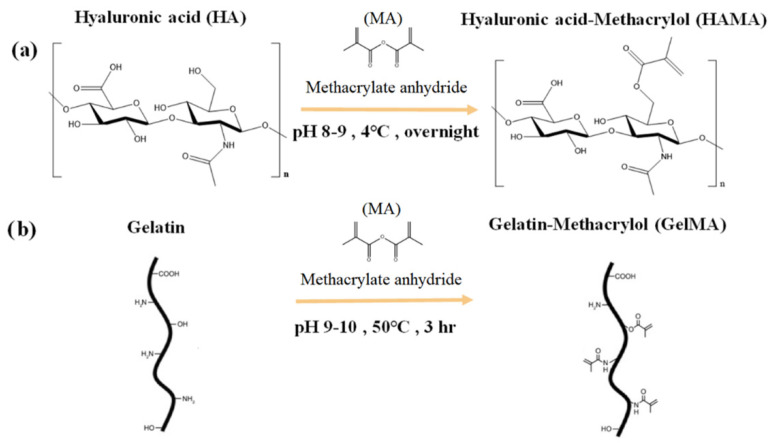
Schematic synthesis of the photo cross–linkable HAMA and GelMA hydrogels. The synthesis of hyaluronic acid methacrylol (HAMA) hydrogel (**a**); The synthesis step of gelatin methacrylol (GelMA) hydrogel (**b**).

**Figure 3 polymers-14-02003-f003:**
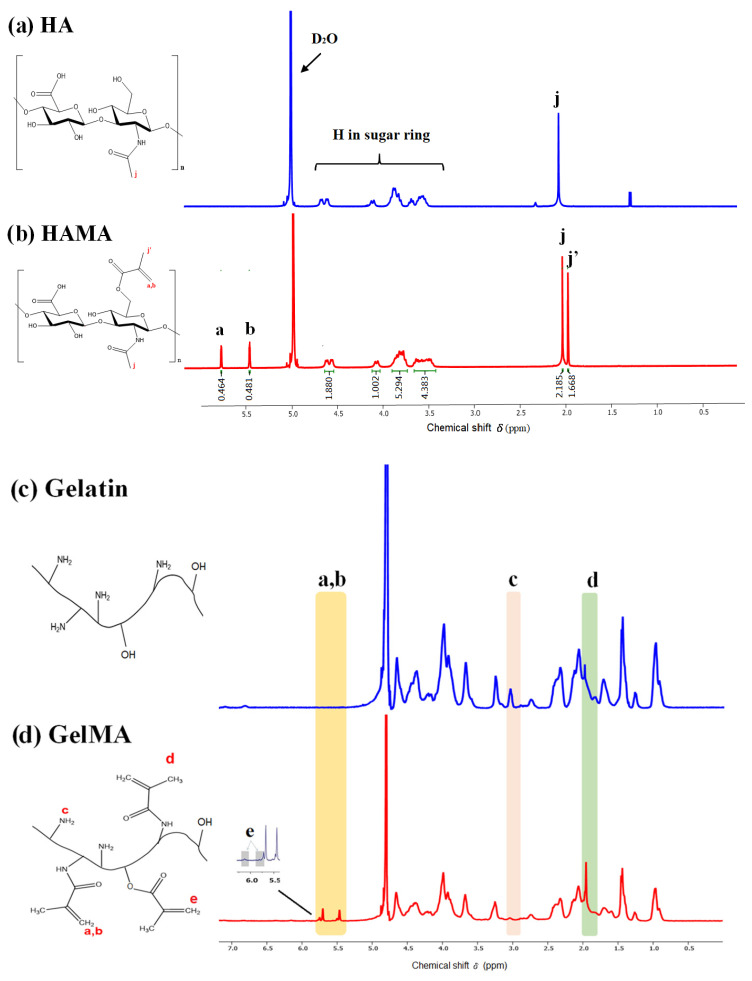
^1^H-NMR spectra of HA (**a**), HAMA (**b**), gelatin (**c**), and GelMA (**d**).

**Figure 4 polymers-14-02003-f004:**
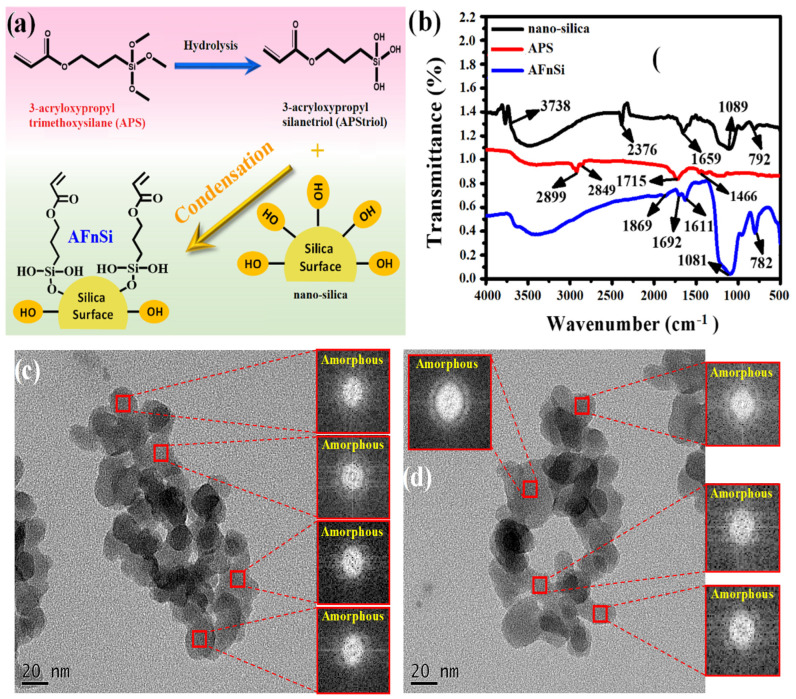
The scheme of AFnSi crosslinker synthesis (**a**); FTIR spectrum of nano–silica, 3–acryloxypropyl trimethoxysilane (APS), and acrylate functionalized nano-silica (AFnSi) (**b**); the TEM images and diffraction patterns of nano-silica (**c**); AFnSi (**d**).

**Figure 5 polymers-14-02003-f005:**
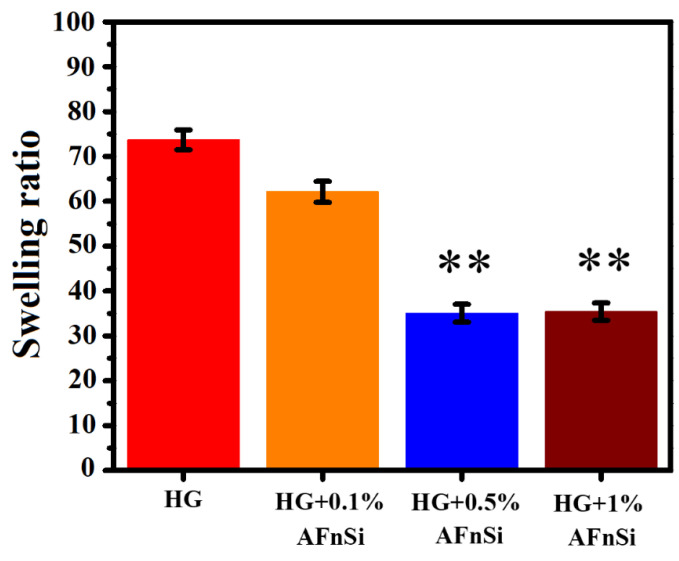
Swelling ratio of HG hydrogel with the different concentrations of AFnSi crosslinker (0, 0.1, 0.5, and 1.0% (*w*/*v*)), respectively. Error bar represents the standard deviation (±; SD), ** *p* < 0.01 compared with HG hybrid hydrogel alone.

**Figure 6 polymers-14-02003-f006:**
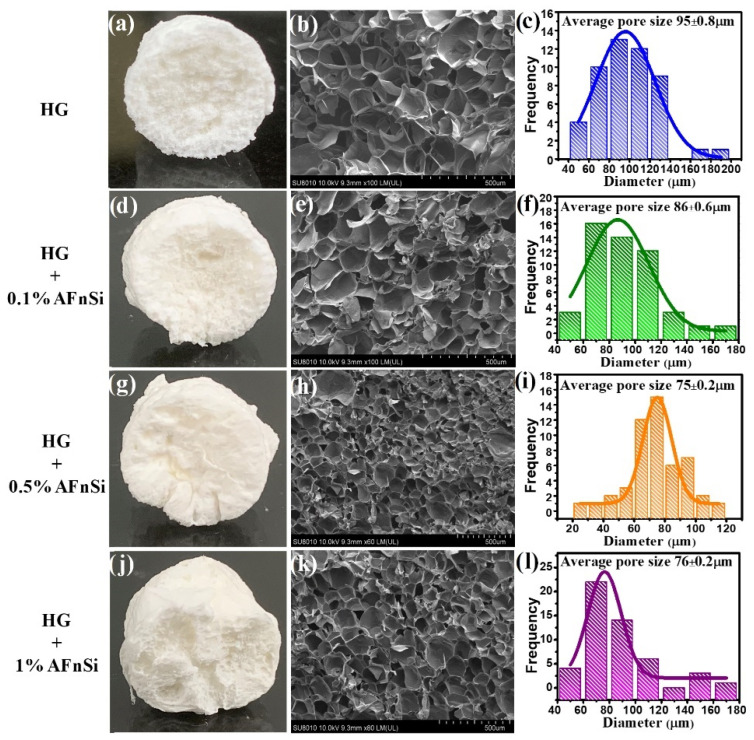
Demonstration of hybrid hydrogel scaffolds with different concentrations of AFnSi, cross-sectional SEM image, and its size distribution histograms. HG (**a**–**c**), HG+0.1% (*w*/*v*) AFnSi (**d**–**f**), HG+0.5% (*w*/*v*) AFnSi (**g**–**i**), and HG+1.0% (*w*/*v*) AFnSi (**j**–**l**), respectively.

**Figure 7 polymers-14-02003-f007:**
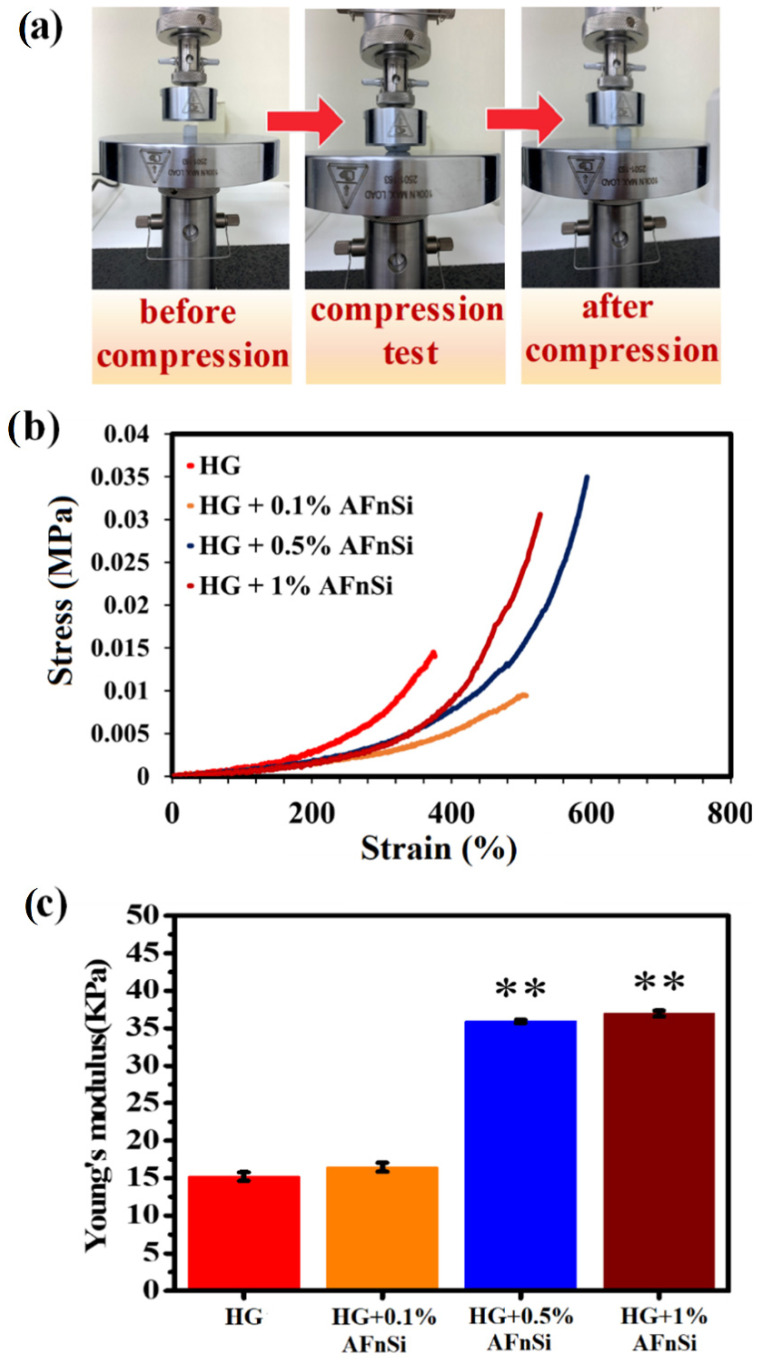
Compression stress-strain curves of HG hybrid hydrogels with the different concentrations of AFnSi crosslinkers (0, 0.1, 0.5, and 1.0% (*w*/*v*)). The schematic photo of the typical compression test process for the HG hybrid hydrogels (**a**) shows the typical curves of compressive stress (kPa) vs. compressive strain (%) (**b**) and their average Young’s modulus (kPa) (**c**). Error bar represents ± SD, ** indicates the significant difference compared to HG and HG/AFnSi values at *p* < 0.01.

**Figure 8 polymers-14-02003-f008:**
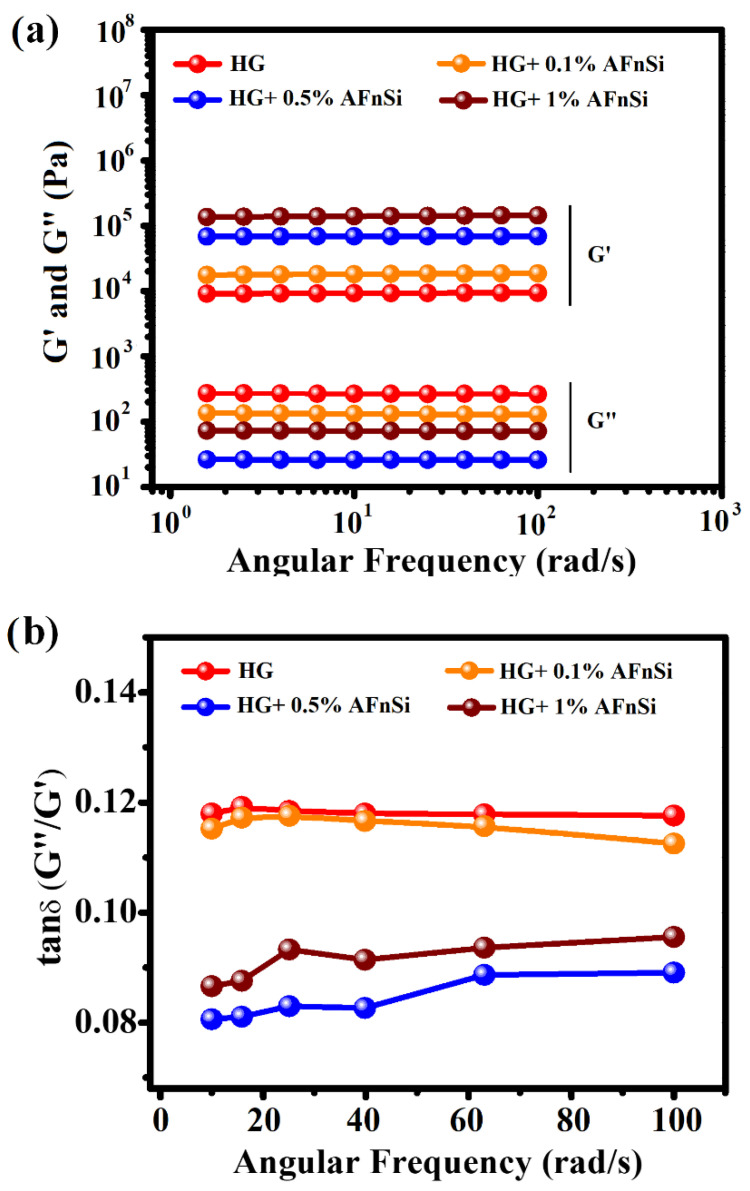
Storage modulus (G′) and loss modulus (G″) (**a**), and loss factor (Tan δ) (**b**), as a function of the angular frequency (ω; rad/s) for the photocrosslinked HG hydrogel with different concentrations of AFnSi crosslinkers such as 0, 0.1, 0.5, and 1.0% (*w*/*v*), respectively.

**Figure 9 polymers-14-02003-f009:**
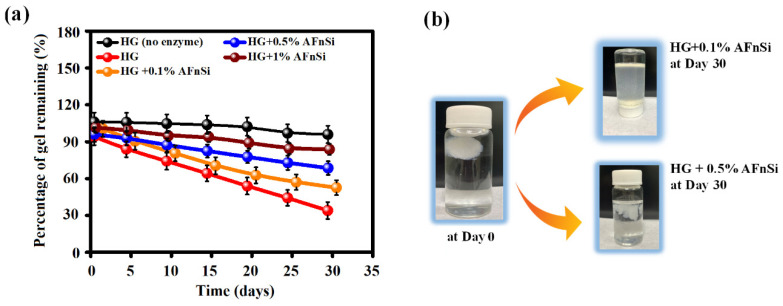
In vitro degradation rate (**a**) of the HG hybrid hydrogel with different concentrations of the AFnSi crosslinker (0, 0.1, 0.5, and 1.0% (*w*/*v*)), respectively, for 30 days by 2.5 U/mL hyaluronidase enzyme. Optical image of hydrogel immersed in PBS with 2.5 U/mL hyaluronidase at day 1 (HG+0.1% AFnSi, HG+0.5% AFnSi immersed in PBS with 2.5U/mL hyaluronidase at day 30 (**b**).

**Figure 10 polymers-14-02003-f010:**
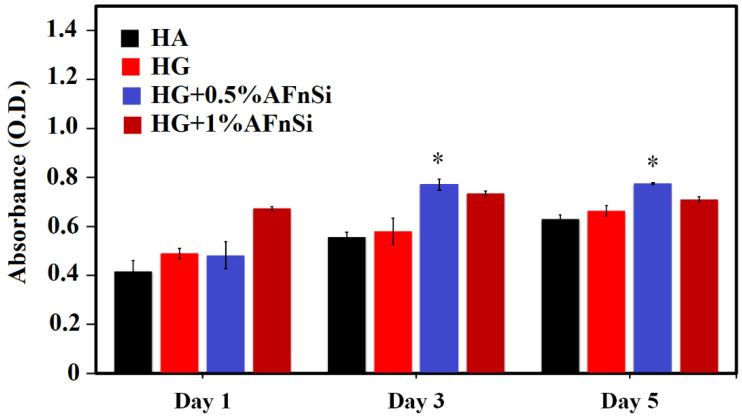
The cell viability evaluation by MTS assay in hADSCs were cultured with the photocured hybrid hydrogel of HG, HG+0.5% AFnSi, and HG+1% AFnSi. Only the HA was controlled. Error bar represents ± SD (*n* = 3); * *p* < 0.05 compared with HA group, respectively.

**Figure 11 polymers-14-02003-f011:**
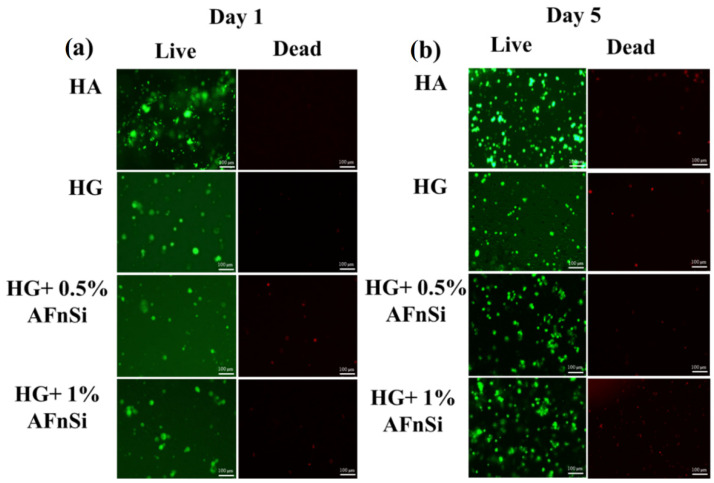
Fluorescence images observation by live and dead staining in hADSCs were cultured with the photocured hybrid hydrogel of HG, HG+0.5% AFnSi, and HG+1% AFnSi on day 1 (**a**) and day 5 (**b**), respectively. HA was used as control. The scale bar represents 100 µm.

**Figure 12 polymers-14-02003-f012:**
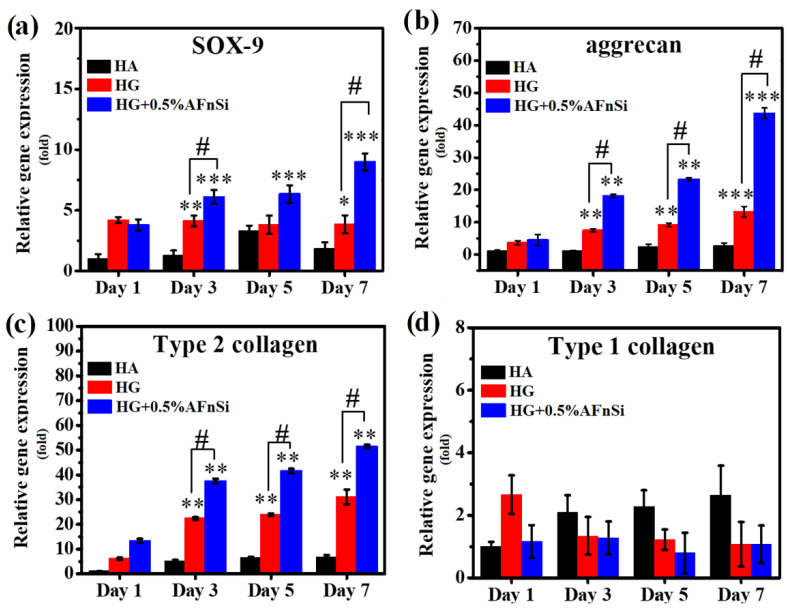
The related chondrogenic/osteogenic gene expression by RT-qPCR analysis in hADSCs were cultured with the photocured hybrid hydrogel of HG and HG+0.5% AFnSi on day 1, day 3, day 5, and day 7, respectively. Only the HA was controlled. Detection of chondrogenic marker gene expression of SOX-9 (**a**), aggrecan (**b**), type II collagen (**c**), and osteogenic marker gene expression of type I collagen (**d**). GAPDH was used as a housekeeping gene. Error bar represents ± SD, *** *p* < 0.001, ** *p* < 0.01, and * *p* < 0.05 compared to HA group, respectively. **^#^**
*p* < 0.05 of HG+0.5% AFnSi compared to HG group.

**Figure 13 polymers-14-02003-f013:**
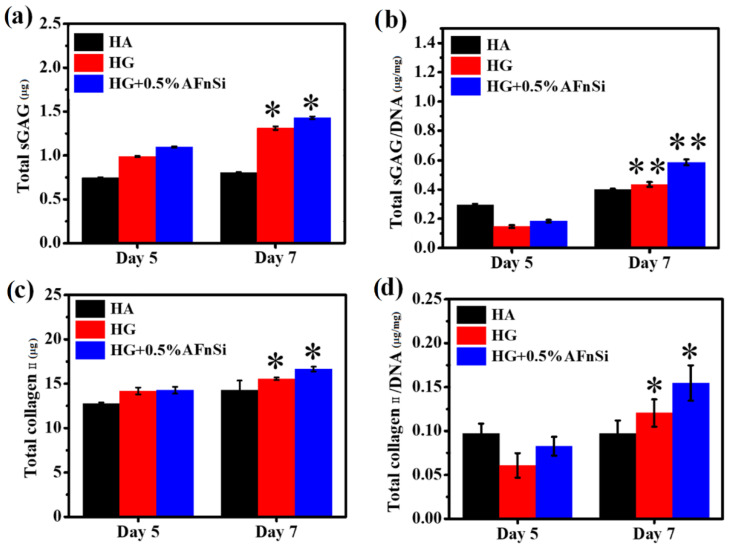
Quantification of sGAG by DMMB assay and collagen type Ⅱ by ELISA method in hADSCs were cultured with the photocured hybrid hydrogel of HG and HG+0.5% AFnSi compared with HA only as of the control on day 5 and day 7, respectively. Total sulfated glycosaminoglycan (sGAGs) (**a**), total sGAG/DNA (**b**) type Ⅱ collagen Ⅱ synthesis (**c**), type Ⅱ collagen/DNA (**d**). Error bar represents ± SD, ** *p* < 0.01, and * *p* < 0.05 compared to the HA group, respectively.

**Table 1 polymers-14-02003-t001:** Reverse and forward primer sequences and annealing temperature are used for RT-PCR.

Gene	The Sequence of Forward and Reverse Primer	Temperature
GADPH	Forward: 5′-TCT CCT CTG ACT TCA ACA GCC AC-3′ Reverse: 3′-CCC TGT TGC TGT AGC CAA ATT C-3′	61 °C
SOX-9	Forward: 5′-CTT CCG CGA CGT GGA CAT-3′ Reverse: 3′-GTT GGG CGG CAG GTA CTG-3′	55 °C
Aggrecan	Forward: 5′-ACA GCT GGG GAC ATT AGT GG-3′ Reverse: 3′-GTG GAA TGC AGA GGT GGT TT-3′	61 °C
Type Ⅱ collagen	Forward: 5′-CAA CAC TGC CAA CGT CCA GAT-3′ Reverse: 5′-TCT TGC AGT GGT AGG TGA TGT TCT-3′	61 °C
Type Ⅰ collagen	Forward: 5′-GGC TCC TGC TCC TCT TAG-3′ Reverse: 5′-CAG TTC TTG GTC TCG TCA C-3′	61 °C

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
