# Peer review of "Characteristic and Chondrogenic Differentiation Analysis of Hybrid Hydrogels Comprised of Hyaluronic Acid Methacryloyl (HAMA), Gelatin Methacryloyl (GelMA), and the Acrylate-Functionalized Nano-Silica Crosslinker"

_polymers, 2022, doi:10.3390/polym14102003_

Round 1
Reviewer 1 Report
In this manuscript, Nedunchezian et al. reported the cartilage tissue engineering application of a unique photo-cured hybrid hydrogel system comprising of hyaluronic acid methacryloyl (HAMA), gelatin methacryloyl (GelMA), and 0~1.0% (W/V) acrylate functionalized nano-silica (AFnSi) crosslinker. The authors examined the physicochemical characteristics of the hybrid hydrogel, and further selected 0.5% (W/V) as the optimum amount of AFnSi. Results showed that the hybrid hydrogel allowed the survival of human adipose-derived stem cells (hADSCs) and can differentiate hADSCs into chondrogenesis. However, the manuscript in the current form still needs major revisions before it is suitable for publication in polymers.
- There are many ambiguities in expression throughout the manuscript, such as the misuse of conjunctions and prepositions and the description of the experimental methods, which greatly reduced the readability and credibility of the manuscript.
- In the part of the Introduction (Line 60), please explain the abbreviations that first appeared in the manuscript and check for similar mistakes in the full text.
- In Figure 1, the description of the experiment process should be put in the annotation of the figure and rearranged the overall structure.
- In Figure 3, the description of the proton nuclear magnetic resonance (1H NMR) spectra by the authors did not agree with the data. There are obvious mistakes in the marking of characteristic peaks in the picture.
- In Section 3.4, the authors said that lyophilization would create artificial pores. To avoid the influence of this process, it is recommended to use a cryogenic scanning electron microscopy (cryo-SEM) for testing.
- In the Abstract, the authors said that the morphological microstructure, mechanical properties, and longer degradation time of the HG+0.5% (w/v) AFnSi hydrogel demonstrated the acellular novel matrix was optimal to support hADSCs differentiation. However, the data in the manuscript showed that there was no significant difference between the 0.5% group and 1% group, it was suggested to adjust the concentration gradient for meaningful screening.
- In this manuscript, there is no uniform specification for drawing figures.
- The recently published review or research articles should be discussed in the revision, for example, 10.1039/D0BM02103B, 10.1016/j.bioactmat.2021.10.002.
Author Response
We thank the reviewer for carefully checking this paper and providing constructive comments. We have amended the manuscript according to the reviewers’ comments and editorial comments; a point-by-point response and a change list are attached to this document and the revised manuscript.

Reviewer 2 Report
The current manuscript described the synthesis and in vitro evaluation of photo-cured hybrid hydrogel system comprising of HAMA, GelMA, and 0~1.0 % (w/v) acrylate functionalized nano-silica (AFnSi) crosslinker. Although the use of nano-silica crosslinker is interesting, the manuscripts contains too many problems that should be carefully addressed.
1,The English expression of the manuscript should be greatly improved. For example, in many parts, the transition words, such as on the other hand or however, are not appropriately used.
2, The introduction part is too long. The authors should make it succinct and more focused. The application of HA and gelatin in cartilage tissue engineering should be discussed more with more relevant reference included such as https://iopscience.iop.org/article/10.1088/1758-5090/ac42de/meta
In the manuscript, it was mentioned that "Because of the hydroxyl groups of nSi surface can be grafted with multiple methacrylation, which promotes the network cross-linking of the 3D environment." However, the authors were not using methacrylated nSi.
Fig.2 showed the 2-methyl-2-propenoic acid as the reactant however in the method section MAA was used.
Fig.7b the x axis value is not correct.
Fig. 8 the authors should use different symbols to distinguish G' and G''.
Why there is no error bar in figure 9?
For the chondrogenic differentiation study, the method section said the cells were cultured in basal medium. Is that accurate?
No scale bar in Fig.11. Also from Fig.11, the cells seemed not spreading on the hydrogel but why? Is gelatin supposed to help the attachment of the cells?
Please provide Safranin O staining on the cells cultured in different hydrogel to support the GAG quantification results.
Author Response

(The authors gave the same response as above.)

Round 2
Reviewer 2 Report
No more comments.